# Generation of human islet cell type-specific identity genesets

Léon van Gurp [1], Leon Fodoulian[2], Daniel Oropeza[1], Kenichiro Furuyama [1,3], Eva Bru-Tari [1], Anh Nguyet Vu[4], John S. Kaddis [4], Iván Rodríguez [2], Fabrizio Thorel[1] & Pedro L. Herrera [1✉]

Generation of surrogate cells with stable functional identities is crucial for developing cell-based therapies. Efforts to produce insulin-secreting replacement cells to treat diabetes require reliable tools to assess islet cellular identity. Here, we conduct a thorough single-cell transcriptomics meta-analysis to identify robustly expressed markers used to build genesets describing the identity of human α-, β-, γ- and δ-cells. These genesets define islet cellular identities better than previously published genesets. We show their efficacy to outline cell identity changes and unravel some of their underlying genetic mechanisms, whether during embryonic pancreas development or in experimental setups aiming at developing glucose-responsive insulin-secreting cells, such as pluripotent stem-cell differentiation or in adult islet cell reprogramming protocols. These islet cell type-specific genesets represent valuable tools that accurately benchmark gain and loss in islet cell identity traits.

[1] Department of Genetic Medicine and Development, Faculty of Medicine, University of Geneva, rue Michel-Servet 1, 1211 Geneva, Switzerland. [2] Department of Genetics and Evolution, Faculty of Sciences, University of Geneva, Quai Ernest-Ansermet 30, 1211 Geneva, Switzerland. [3] Center for iPS Cell Research and Application (CiRA), Kyoto University, 53 Shogoin-Kawahara, Sakyo, 606-8507 Kyoto, Japan. [4] Department of Diabetes & Cancer Discovery Science, Arthur Riggs Diabetes and Metabolism Research Institute, Beckman Research Institute, City of Hope, 1500 E Duarte Rd, Duarte, CA 91010, USA. ✉email: pedro.herrera@unige.ch

Blood glucose homoeostasis is regulated mainly by the pancreatic "islets of Langerhans", which are formed by several types of endocrine cells. The bulk of these pancreatic islets is made up of α- and β-cells, defined by the release of glucagon and insulin, respectively, upon blood glucose level variations. Three other islet cell types, namely γ-, δ- and ε-cells, secrete the hormones pancreatic polypeptide, somatostatin and ghrelin, respectively, which contribute to the regulation of α- and β-cell secretory activities.

The ontogeny of the different islet cell types has been largely studied in mice, where endocrine progenitors located in the ductal epithelium start expressing either ARX or PAX4, pushing their specification towards α- or β-cell fates[1]. Cell type-specific transcription factors then guide cells towards maturation. For α-cells, these include MAFB, IRX2 and BRN4, while PDX1, MAFA and NKX6.1 are key factors for β-cell maturation. Other transcription factors, like RFX6, PAX6, FOXO1, FOXA1, FOXA2 and NEUROD1 seem to play a more ubiquitous role during islet cell type specification[2]. In adult cells, cellular identity is largely defined by cell type-specific transcriptional networks regulating cellular functions. For β-cells, the transcriptional landscape associated with glucose sensing, calcium and incretin signalling, insulin processing and secretion, is relatively well defined[3]. Much less is known about the functional and transcriptional regulation in the other cell types, particularly in human islets.

The development of single-cell RNA sequencing (scRNA-seq)[4,5], a technique allowing the profiling of the mRNA content of individual cells from heterogeneous tissues, has offered an opportunity to better characterize the cellular identities of the different human pancreatic islet cell types. Single-cell sequencing has progressively become more standardized[6,7], but during the process, there have been a high diversity in cell capture techniques, library preparations, sequencing protocols and data processing tools. Thus, despite expecting relatively similar results, significant differences were observed, making side-by-side comparisons complex. For example, from the 171 genes that were detected to be upregulated in T2DM β-cells, only one was found in more than one dataset[8]. Equally, when comparing β-cell heterogeneity results, only six out of 43 genes were shared by at least three out of five datasets, and none were shared by all[9]. This lack of overlap complicates the robust definition of islet cellular identities.

To get a more comprehensive view of what defines cell type identities, geneset analysis tools have proven to be highly valuable[10–12]. For example, by assessing sets of genes, information can be gained on activated or repressed signalling pathways involved in biological processes and/or molecular functions[13]. Unfortunately, specific genesets defining the identity of the different islet cell types are rare or do not exist. The current ones do not necessarily reflect the identity of the mature functional cells as they often include developmental or imprecise markers. For example, the widely used Pancreas Beta Cells geneset from the Hallmark collection[14] contains non-β genes like GCG, MAFB, PCSK2 and SST, or genes that are strongly regulated during endocrine specification, like NEUROG3 and NKX2-2.

Here, we have performed a meta-analysis aimed at defining the transcriptomic identity profiles for the different islet cell types through a common standardized pipeline. Previous reviews had focused on overlap between datasets in the context of type 2 diabetes and β cell heterogeneity[8,9]. After validating their efficacy by determining the optimal trade-off between geneset sensitivity and specificity, we use these genesets to assess identity plasticity in a range of different settings, from (trans)differentiation to the impact of the disease. The genesets we generate here can be downloaded from the Molecular Signatures Database (https://www.gsea-msigdb.org/gsea/msigdb), or directly applied in our web-app scPancMeta (https://rapps.hirnetwork.org/scPancMeta).

## Results

### In silico purification of human mono-hormonal α-, β-, γ- and δ-cells.
We conducted a meta-analysis studying in silico purified populations of singlet mono-hormonal α-, β-, γ, and δ-cells from seven published human islet single-cell transcriptomic datasets (54 donors in total; Tables S1, S2)[15–21], with the goal to generate cell type-specific genesets. These key identity genes should consistently and robustly be expressed across the different independent datasets for each islet cell type. Raw sequencing reads from seven datasets were first downloaded and mapped against the GRCh38 transcriptome using Kallisto[22]. Individual datasets were independently processed using Seurat[23]. Cell types were inferred based primarily on unsupervised clustering, after which corrections were made based on hormone expression (see Methods; Fig. S1A; Table S3).

First, we aimed at identifying and excluding cells that represent doublets as they may impact on the identification of relevant islet cell genes. We decided to use two independent predictive tools (Scrublet and DoubletFinder) to maximize the probability of identifying doublets[24,25]. The amount of overlap between the tools, and the localization of doublets was highly variable between datasets (Fig. S1B; Table S3). This variability has been described previously in the context of yet another doublet removal tool (DoubletDecon)[26], and justifies the use of more than one tool to be more stringent in the removal of doublets. Regardless, all cells identified as putative doublets by either tool were removed in order to minimize downstream interference.

Next, we explored overall islet composition based on final cell type allocation (Fig. S1C). We found that α-cells were the most abundant cell type, constituting 58.9% of all NDM islet cells. The β-, γ- and δ-cells only comprised 30.4%, 4.4% and 6.1% of the NDM islet cells, respectively. Compared to what was found in situ by immunofluorescence and CyTOF[27–29], the β/α-ratio was generally lower in scRNA-seq studies[15–21]. This β/α-cell ratio is known to be impacted by islet isolation[30], and this may be even more so by dispersing islets into single cells. Similar compositional results were observed in T2DM donors (Fig. S1C). We then aimed to identify bi-hormonal cells that were still detected after doublet removal. As previously reported[31], the abundance of bi-hormonal cells seemed slightly elevated in T2D donors, but in our analysis, this difference was not significant (Fig. S1D).

Despite limited numbers, we tried to resolve the transcriptomic signature of GCG/INS bi-hormonal cells in the human pancreas. We found GCG/INS bi-hormonal cells in both α- and β-cell clusters and calculated differentially expressed genes compared to their mono-hormonal counterparts (Fig. S2A). Interestingly, bi-hormonal cells only had a small set of genes upregulated compared to mono-hormonal cells, and none downregulated, thus reflecting their hybrid phenotype, which retains the genes linked to the mono-hormonal cell type while acquiring additional identity genes from the alternative cell type. Bi-hormonal cells in the α-cluster upregulated key genes related to β-cell identities, like INS, IAPP, DLK1 and RBP4. Inversely, bi-hormonal cells in the β-cell cluster upregulate α-cell-related genes like GCG, ARX, FEV and TM4SF4 (Fig. S2A). Expression patterns for these genes were comparable between different datasets (Fig. S2B).

After these analyses, all cells originating from T2D donors, and all bi-/poly-hormonal cells, were excluded from downstream analyses, which were thus performed on pure populations of singlet mono-hormonal α-, β-, γ- and δ-cells from non-diabetic donors.

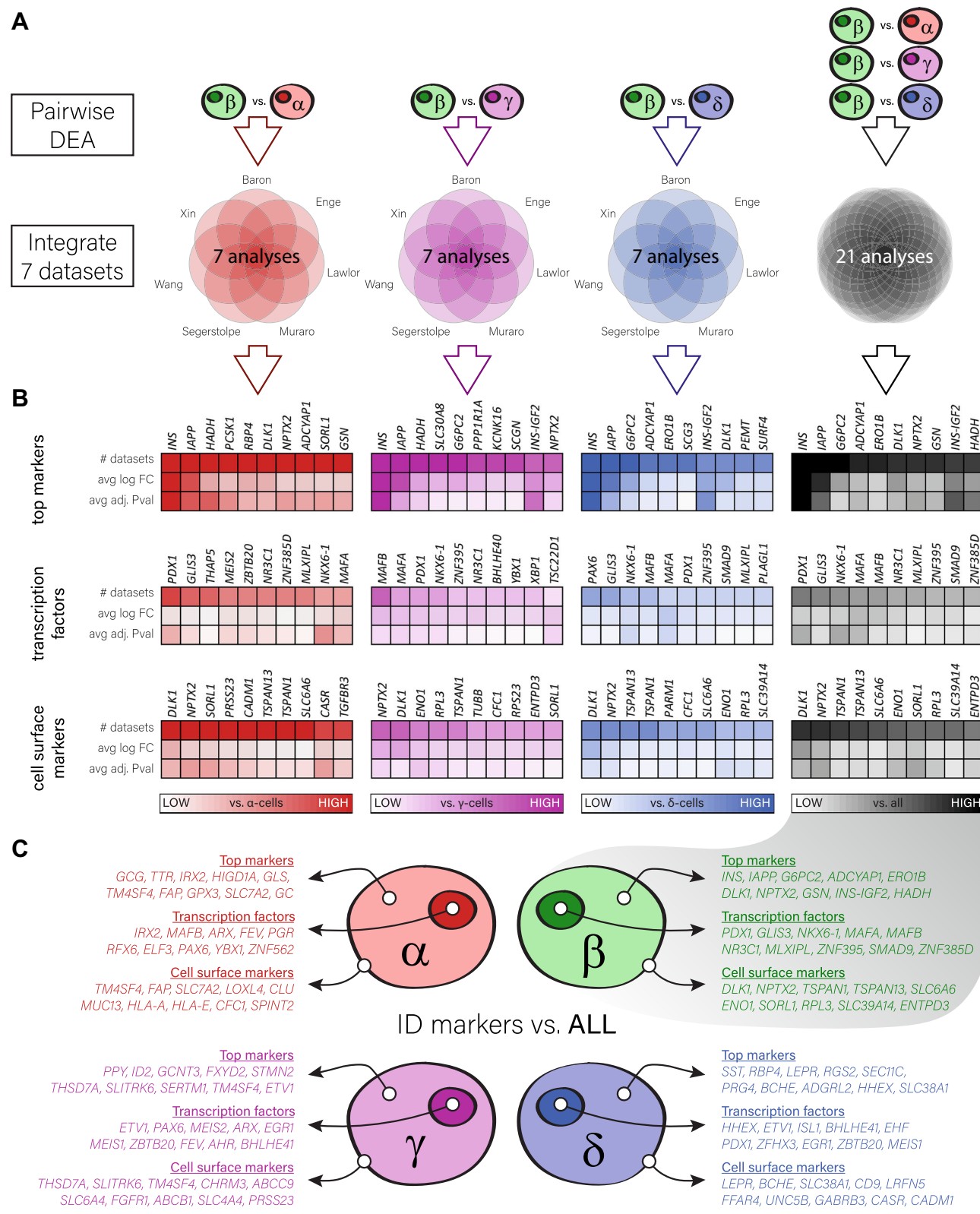

**Defining identity genes for the human pancreatic α-, β-, γ- and δ-cells.** Next, in order to compare each islet cell type directly with the other cell types, we made pairwise comparisons. For example, we compared β-cells first to α-, then to γ- and δ-cells in all seven datasets (Fig. 1A, in red, magenta and blue). The differentially expressed genes we found compared to specific cell types were then integrated, and identified genes were organized based on the number of analyses they were found in, and the average log fold

change and adjusted p-value between all analyses. Then, we combined all pairwise analyses together to elucidate the overall transcriptomic profile of each cell type (Fig. 1A, in black). For this, we integrated all analyses (vs. three different cell types in seven datasets; 21 analyses in total), and organized the results in the same way as described above. This was done similarly for each of the major islet cell types (α-, β-, γ- and δ-cells). (Fig. 1B, Fig. S3A, and Supplementary data 1–4). Of note, we found that all

**Fig. 1 Generation of islet cell type-specific genesets by intersecting differentially expressed genes from independent single-cell transcriptomics datasets. A** Differential expression was calculated in seven independent datasets in a pair-wise manner between all cell types (α vs. β, α vs. γ, α vs. δ, β vs. γ, β vs. δ, γ vs. δ) using either the negbinom test on UMI based data (Baron, Muraro) or the MAST test. Then, data from all seven datasets were integrated for a direct comparison between cell types, and in a combined manner to elucidate general cell type-specific identity genes. **B** Top differentially expressed genes, top transcription factors and top genes that encode cell-surface proteins that characterize β-cells, in direct comparisons to α-, γ- and δ-cells (in red, magenta and blue, respectively), and in a combined manner to define general β-cell identity genes (in black). Genes were ordered first on the number of analyses in which they were found to be differentially expressed, then based on a rank score that comprised both the Bonferroni corrected p-value and the log fold-change. Darker colours indicate a higher number of analyses, a higher log fold-change or a more significant adjusted p-value. **C** An overview of the top 10 identity markers, top 10 transcription factors and top 10 cell surface encoding genes, per cell type. α-cell identity genes in red, β-cell identity genes in green, γ-cell identity genes in magenta and δ-cell identity genes in blue. Source data are provided in the supplemental tables and as a source data file.

datasets do contribute to the composed identity genesets, with a certain bias that indicated that some datasets (Muraro, Segerstolpe) typically contributed more than the others (Fig. S3B). We do not observe a confounding bias based on the technique used to generate the datasets[17–19,21]. Also, we do not observe obvious contradictory results, where genes are differentially expressed in different directions in different datasets (Supplementary data 1–4).

In order to test the validity of the gene list content obtained from our meta-analysis, we first reviewed the literature (Supplementary table 4) to assess the extent of genes already assigned to a given cell type. We found many undisputable markers for α- and β-cells amongst the 20 top regulated genes (14 and 16 genes with specificity and/or a known function in α- and β-cells, respectively; Supplementary table 4). For example, *G6PC2*, *ADCYAP1* and *HADH* in β-cells all have been demonstrated to be involved in maintaining glycemia[32–34]. Amongst the top regulated transcription factors, key identity regulators *PDX1*, *NKX6-1* and *MAFA* are highly regulated, while *ENTPD3* is one of the top regulated cell surface markers[32]. In α-cells, *TTR*[35] and *SLC7A2*[36] are amongst the top regulated genes. *IRX2*[33] is recognized as the most strongly regulated transcription factor, while *TM4SF4*[18] and *FAP*[37] are the most highly regulated cell surface markers (Fig. 1C, Fig. S3A). The γ- and δ-cells were much more poorly characterized, but we did find here a number of identity markers for each of these cell types as well. For γ-cells, these included the transcription factors *ETV1*[38] *PAX6*[39] and *ARX*[40], where *HHEX*[41], *ETV1*[38] and *ISL1*[42] were found in δ-cells. Other interesting top markers for γ-cells included *FXYD2*[43] and *TM4SF4*[18], and for δ-cells *CD9*[44], *LEPR*[45] and *FFAR4*[46] (Fig. 1C, Fig. S3A).

Next, to further strengthen the validity of the identity genes we here describe, we aimed to validate an extra number of genes using single-molecule fluorescent in situ hybridization (smFISH). This technique allowed us to multiplex measure relative gene expression with spatial resolution in human formalin-fixed, paraffin-embedded pancreas sections. We thus confirmed that *IAPP* and *GSN* have higher expression in β- than α-cells, while *TTR*, *GLS*, *SPINT2* and *SERPINA1* were more highly expressed in α- than β-cells (Fig. S4A–C). Finally, we found that *ARX*, a well-known α-cell marker, was found to be equally well regulated in γ-cells as in α-cells at mRNA levels (Fig. S3B, Supplementary data 1 and 3). We validated the presence of ARX in both adult α- and γ-cells of the human pancreas at the protein level (Fig. S4D).

**Generation of a single-cell transcriptomics dataset enriched for γ-, δ- and ε-cells.** We next aimed to better define the less common cell types in the pancreatic islets in more detail. To that effect, we generated a dataset containing single-cell transcriptome profiles of human pancreatic islets that were enriched for the γ-, δ- and ε-cell fractions. To do this, we optimized our previously reported cell sorting protocol (Fig. 2A)[47]. In short, dissociated human pancreatic islet cells were antibody-labelled, as described previously[44,48]. The enriched γ-, δ- and ε-cell

populations were collected by flow cytometry (FACS) to generate single-cell libraries.

The dataset was generated using 10× Genomics single-cell gene expression kit and processed using the analysis pipeline described above. It contained over 15,000 cells after quality control filtering and doublet removal. Cell types were inferred as previously, primarily based on unsupervised clustering. In total, we defined 2492 cells (16.1%) as α-cells, 2767 cells (17.9%) as β-cells, 4082 cells (26.4%) as γ-cells, 2315 cells (14.9%) as δ-cells and 666 cells (4.3%) as ε-cells. We also found 401 bi-hormonal cells after doublet removal (2.6%), most of which co-expressed *INS* and *SST*. The distribution between islet cell types in our dataset was thus much more uniform than in previously generated datasets (Fig. 2B) confirming enrichment of the γ-, δ- and ε-cell fractions. The remaining 2862 cells (18.5%) represent non-endocrine types such as ductal, acinar, stellate and immune cells (Fig. 2C; Supplementary table 5). Cells originating from different donors clustered based on cell type, not on donor identity (Fig. S5A). As median values, cells expressed 10,557 UMIs and 2958 genes (Fig. S5B). Cell type-specific clusters only expressed one single hormone, matching their identity (Fig. S5C). As a control, we integrated our γδε-enriched dataset with the Baron and Muraro datasets, which were also UMI based, to validate that the cells we obtained represented the same populations as those obtained during unbiased cell selection (Fig. S5D).

We calculated the differential gene expression between different islet cell types to determine cell type-specific identity genes. We found *TTR*, *GPX3*, *SLC7A2* and *FAP* in α-cells, which were also in the above top 10 α-cell identity gene list, besides well-known transcription factors like *IRX2* and *MAFB* (Figs. 1C, 2D, Supplementary data 1 and 5). Regarding β-cells, we found top markers like *ADCYAP1* and *HADH*, besides well-known transcription factors like *PDX1*, *MAFA* and *NKX6-1* (Figs. 1B–C, 2D, Supplementary data 2 and 5). In γ-cells, top markers like *STMN2* and *SERTM1* were retrieved, just like previously validated transcription factors like *PAX6* and *ETV1*, while δ-cells expressed top markers like *RGS2*, *SLC38A1* and *SEC11C*, together with transcription factor *HHEX* (Figs. 1C, 2D, Supplementary data 3, 4 and 5). Previously described ε-cell-specific genes, like *SPINK1*, *APOH*, *ASCL1* and *FRZB*[49], were expressed in our dataset as well. Interestingly, we also found α- and γ-cell-related markers, such as *F10* and *ETV1*, to be strongly expressed in ε-cells. Particularly, the expression of *ARX*, a well-established key regulator of α-cell identity[50,51], was found to be more highly expressed in both γ- and ε-cells, and we detected it on the protein level in γ-cells as well (Fig. 2D; Fig. S4D; Supplementary data 5).

In summary, we have generated a single-cell transcriptome dataset of human pancreatic islets containing thousands of α-, β-, γ- and δ-cells, and hundreds of ε-cells, a dataset that thus provides an unusual characterization of the less common human islet cell types. This dataset will be referred to as the γδε-enriched dataset from here on.

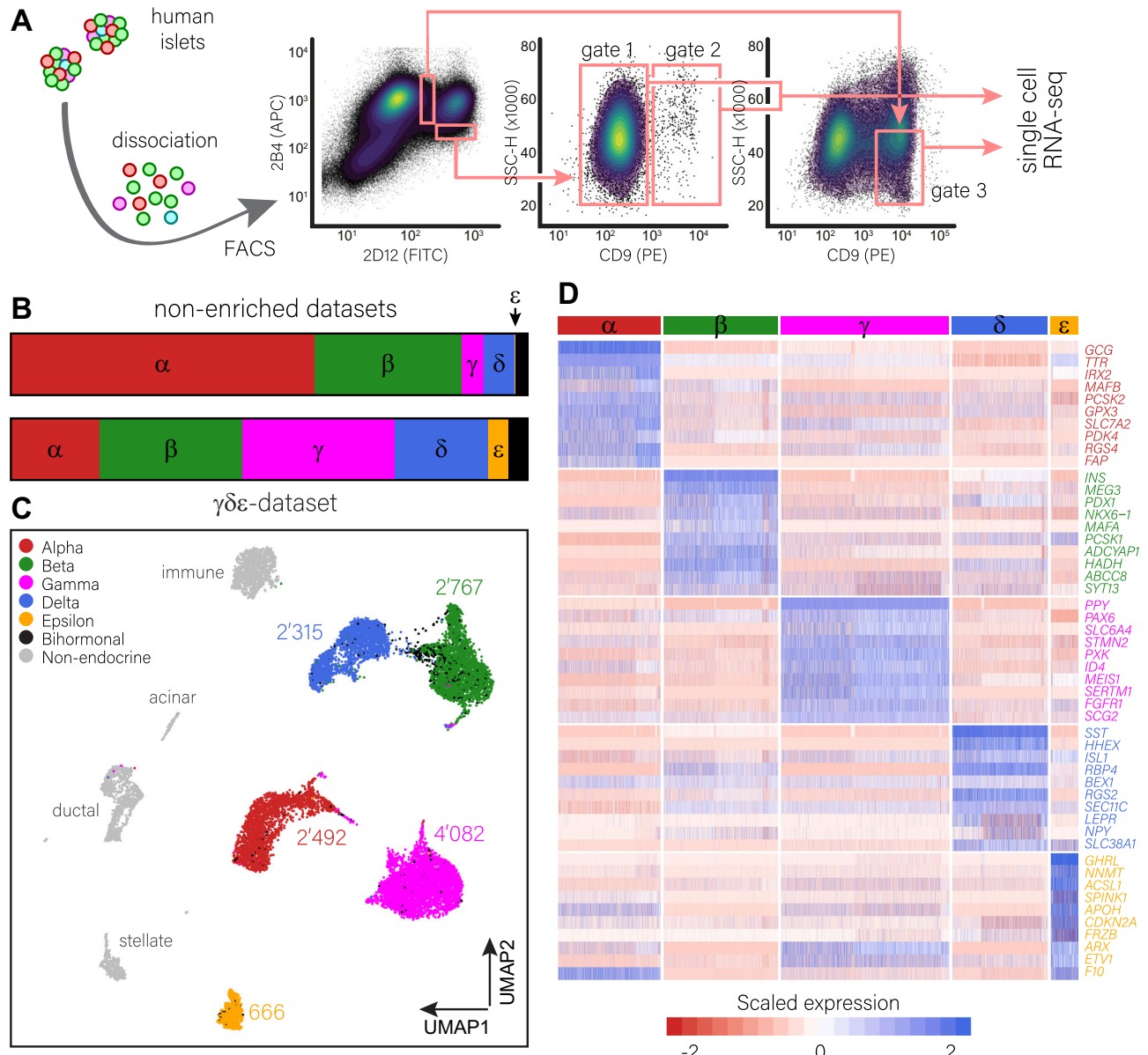

**Fig. 2 Generation and validation of a single-cell transcriptomics dataset enriched for γ-, δ- and ε-cells. A** Strategy for the enrichment of human γ-, δ- and ε-cells. Dissociated human islets were labelled with cell-surface antibodies and the different fractions were processed and collected as described. Cells from gate 2 were complemented with cells from gate 1 to 15,000 cells (γ/ε fraction). The δ-fraction of 15,000 cells was collected in gate 3. **B** Compared to the unsupervised islet cell collection used to produce non-enriched datasets, our dataset contains smaller fractions of α- and β-cells but larger fractions of γ-, δ- and ε-cells. **C** UMAP dimensional reduction representation of the final dataset. Cells are colour-coded based on their identity. Populations of α-, β-, γ- and δ- cells each contain thousands of cells, while the ε-cell fraction contains hundreds of cells. **D** Heatmap showing a representative selection of manually selected identity markers for each of the cell types. Low expression levels are marked in red, high expression in blue. The colour bar above indicates the specific populations: α-cells (red), β-cells (green), γ-cells (magenta), δ-cells (blue) and ε-cells (orange). Source data are provided in the supplemental tables and as a source data file.

**Generation of optimal genesets that accurately define islet cell type-specific identity**. We next aimed to define a set of cell type-specific genesets, based on the identity genes generated by integrating differential expression from the seven published datasets. For each islet cell type, differential expression was analysed against the three other cell types in a pairwise manner, resulting in a total of 21 integrated analyses, as shown in Fig. 1 (black heatmaps). Of note, genes that were found in more analyses were considered to be more predictive of cellular identity.

For example, three genes were consistently found in all of the 21 β-cell analyses: *INS*, *IAPP* and *G6PC2*. While their combination would make a very specific β-cell identity geneset, the

sensitivity of such a geneset would be low as many other genes describing β-cell identity are missing. Conversely, 1872 genes would be found as β-cell identity genes if the selection criteria were defined as a "marker present in at least one" of the 21 analyses. Such a geneset would be very sensitive, as all genes describing β-cell identity would likely be included, but with very low specificity, for it would also contain many exotic genes with low predictive value for β-cell identity.

Here, we aimed at defining the optimal trade-off between specificity and sensitivity, to produce genesets with as many highly predictive identity genes as possible, while removing as many genes with low predictive value as possible. To do this, we

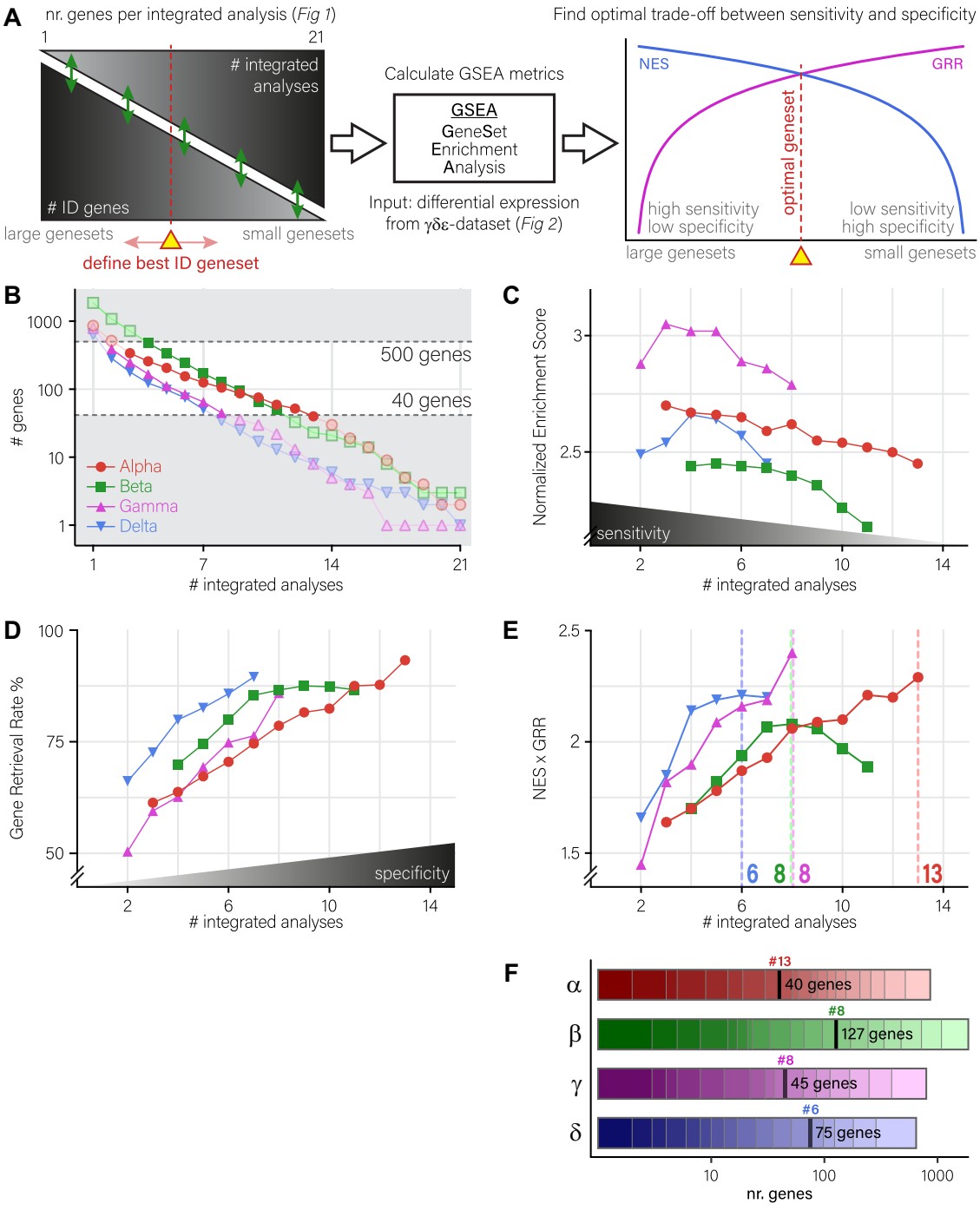

**Fig. 3 Generation of islet cell type-specific identity genesets. A** Methodology to determine the optimal genesets from our lists of identity genes. First, 21 incrementally smaller genesets were generated per cell type, including genes with an increasingly higher number of integrated analyses. To define the best identity geneset, we used GSEA to generate normalized enrichment scores (NES; proxy for geneset sensitivity) and gene retrieval rates (GRR; proxy for geneset specificity) based on differential expression in our γδε-dataset. The optimal geneset was defined as the geneset with the highest combined NES/GRR metrics. **B** Geneset sizes for each cell type, regarding every possible intersect level. Genesets were filtered to contain between 40 and 500 genes (darker colours). Genesets outside this range (lighter colours) were not considered for downstream evaluation. **C** Mean sensitivity score (normalized enrichment; $n = 3$ independent experiments) for each geneset for each cell type. **D** Mean specificity score (gene retrieval rate; $n = 3$ independent experiments) for each geneset for each cell type. **E** Multiplication of NES and GRR scores ($n = 3$ independent experiments) for each geneset for each cell type. Per cell type, the highest measured value is indicated by a dotted line, and the appropriate number of integrated analyses is indicated on the x-axis. Genesets were defined to include all genes with at least this amount of integrated analyses, resulting in genesets sizes as indicated in the top left corner. **F** Determination of final geneset sizes by applying the determined # integrated analyses on the lists of ID genes generated in Fig. 1. For each cell type, the determined cut-off is indicted as a thick black line, numbers indicated with a # indicate the cut-off value determined in panel (**E**). α-, β-, γ- and δ-cell genesets in red, green, magenta and blue, respectively. Source data are provided as a source data file.

needed to determine what would be the minimal number of integrated analyses a gene should be found in, in order to be included in the geneset (Fig. 3A, left part). To that effect, we made use of GeneSet Enrichment Analysis (GSEA)[10]. Output from GSEA includes the Normalized Enrichment Score, which represents the correlation between the geneset and the observed phenotype (NES; proxy for sensitivity), and the Gene Retrieval Rate, which represents the fraction of observed genes from a geneset during analysis (GRR; proxy for specificity). Thus, the number of intersected analyses where the relation between NES and GRR is the highest would represent the ideal geneset for each of the different islet cell types (Fig. 3A, right panel).

For each cell type, the number of genes per geneset was inversely correlated with the number of integrated analyses they were found in. In general terms, α- and β-cell identity genesets contained more genes at any given intersect level than γ- and δ-cell identity genesets (Fig. 3B). For downstream analyses, we excluded any geneset that contained more than 500 or less than 40 genes, due to default thresholding settings during GSEA (Fig. 3B; see Methods). For the remaining genesets, we calculated both NES and GRR in each pairwise comparison, and used the mean value as a measure. While the NES (sensitivity) tended to decline with the number of intersect levels (Fig. 3C), we observed a positive correlation between the GRR and an increase in the number of integrated analyses, indicating that the genesets were indeed becoming more specific for each cell-type (Fig. 3D). The optimal trade-off between specificity and sensitivity was determined as the highest value of the multiplication between the NES and GRR (Fig. 3E).

This resulted in four genesets, one for each cell type, that consisted of 40, 127, 45 and 75 genes for α-, β-, γ- and δ-cells, respectively (Fig. S6A). Of these genes, some were predictive of more than one cell type. Most shared genes (10) were found between β- and δ-cells: *NPTX2*, *HADH*, *PCSK1*, *TIMP2*, *SORL1*, *RBP4*, *CADM1*, *DHRS2*, *SCD5* and *CASR*. δ- and γ- cells next shared 7 genes: *ETV1*, *ABCC9*, *AQP3*, *CALB1*, *DPYSL3*, *CPB1* and *AKAP12*. α- and γ- cells shared 3 genes (*TM4SF4*, *TMEM176B* and *GC*), α- and δ-cells 2 genes (*PAPPA2* and *GPX3*), β- and γ-cells 1 gene (*FXYD2*), and α- and β-cells also share 1 gene (*PEMT*). The remaining genes (34, 115, 34 and 56 genes for α-, β-, γ- and δ-cells, respectively) were only predictive for a single cell type (Fig. S6B).

**Islet cell type-specific genesets perform better on combined sensitivity and specificity than previously published cell type-specific lists of identity genes.** In order to ascertain if our genesets represent a significant gain over previously published lists, we decided to perform an in-depth comparison with previously published lists of identity genes from the original manuscripts (Lawlor, Muraro, Segerstolpe and Xin datasets)[17–19,21]. For β-cells, we also included two genesets from the Hallmark[14] and Reactome[12] collections that have been commonly used to evaluate β-cell identity. For evaluation purposes, we analyzed all these genesets by GSEA using two more recent datasets[52,53] besides our γδε-enriched dataset. This way, each geneset could be assessed nine times (in three datasets, in a pairwise manner against three other cell types).

Between these different genesets, each originating from specific datasets, thus unique in their composition, we still found general trends indicating that larger genesets were both more sensitive and less specific (Fig S7). We found that our proposed genesets always performed equally well or better than genesets from the original datasets, both regarding the normalized enrichment (sensitivity) and gene retrieval (specificity; Fig. 4). More importantly, our genesets are the only genesets that combine high sensitivity and specificity, while the previous from the Muraro and Segerstolpe datasets suffer on specificity and those from the Lawlor and Xin datasets suffer on sensitivity (Fig. 4C).

Equally important, our genesets never failed GSEA analyses under control conditions, unlike the γ- and δ-cell identity genesets from the Lawlor and Xin datasets, which failed in all conditions (Fig. 4D). For β-cells, we also assessed how well our β-cell identity genesets performed compared to well-known genesets from the Hallmark (Pancreas beta cells) and Reactome (Regulation of gene expression in beta cells) databases. The Hallmark geneset failed in five out of nine conditions, and generated lower NES and GRR scores for the conditions that did work, while the Reactome genesets failed in all conditions. These genesets thus seem poorly appropriate choices to evaluate the transcriptomic signature of pancreatic β-cells.

**Islet cell type-specific genesets help evaluate and characterize dynamic changes in islet cell identity during differentiation, conversion and in disease.** Substantial efforts are currently made by many laboratories to generate functional β-like cells for future diabetes therapies. We took advantage of the genesets obtained from our meta-analysis to evaluate the progression in identity toward the α- and β-cell phenotype in ES/iPS differentiation and adult islet cell reprogramming protocols. Likewise, we applied our geneset to evaluate identity changes caused by type 2 diabetes.

Before assessing iPS/ES differentiation, we first tested whether our genesets were suitable to evaluate natural α- and β-cell differentiation during pancreas development, using a published scRNA-seq dataset enriched for mouse pancreatic cells progressing towards an endocrine cell fate[54]. For this purpose, genes in the human genesets were converted to their mouse orthologs (using Ensembl reference genome GRCm38 build 100; Supplementary data 7). Some genes were translated into more than one mouse gene (like *Ins1* and *Ins2* for the human *INS* gene), while others did not translate in a mouse orthologue (like the *INS-IGF2* read-through in beta cells, and *C22orf42* in delta cells). The resulting mouse genesets contain 43, 126, 45 and 75 genes for α-, β-, γ- and δ-cells, respectively, compared to 40, 127, 45 and 75 genes for α-, β-, γ- and δ-cells in the human genesets (Supplementary data 6 and 7).

We then determined how well these mouse orthologous genesets function to evaluate dynamic processes such as α- and β-cell differentiation. After dataset downloading, we calculated the differentially expressed genes between the different cell types as defined by clustering in the original manuscript, and used the rank files generated from these analyses as input for GSEA. No changes in islet cell identity were found for cells progressing from the ductal epithelium towards *Neurog3* expression, nor from *Neurog3* expressing endocrine precursors towards *Fev* expressing cells (Fig. 5A). After this, cells differentiating towards the α-cell phenotype had enrichment only in α-cell identity genes. Inversely, cells progressing towards the β-cell fate gained β-cell identity genes exclusively (Fig. 5A). We also evaluated how well the genesets of the original manuscripts were able to predict these identity changes. Not all of these were able to detect enrichment for endocrine progenitors developing towards either α- or β-cells, and the ones that did detect it yielded lower normalized enrichment scores than our genesets (Supplementary table 6). In conclusion, our genesets display both good sensitivity and specificity in this context as well.

Next, we investigated how well ES/iPS protocols mimic the in vivo process of α- and β-cell differentiation, and if our genesets can help evaluating differentiation protocols. To this end, we downloaded recently published single-cell transcriptomic datasets in which the cells in different culture stages were investigated from stage 3 to stage 6[55]. Transcriptomics data was published for two culture protocols to assess β-cell maturation (protocol x1 and

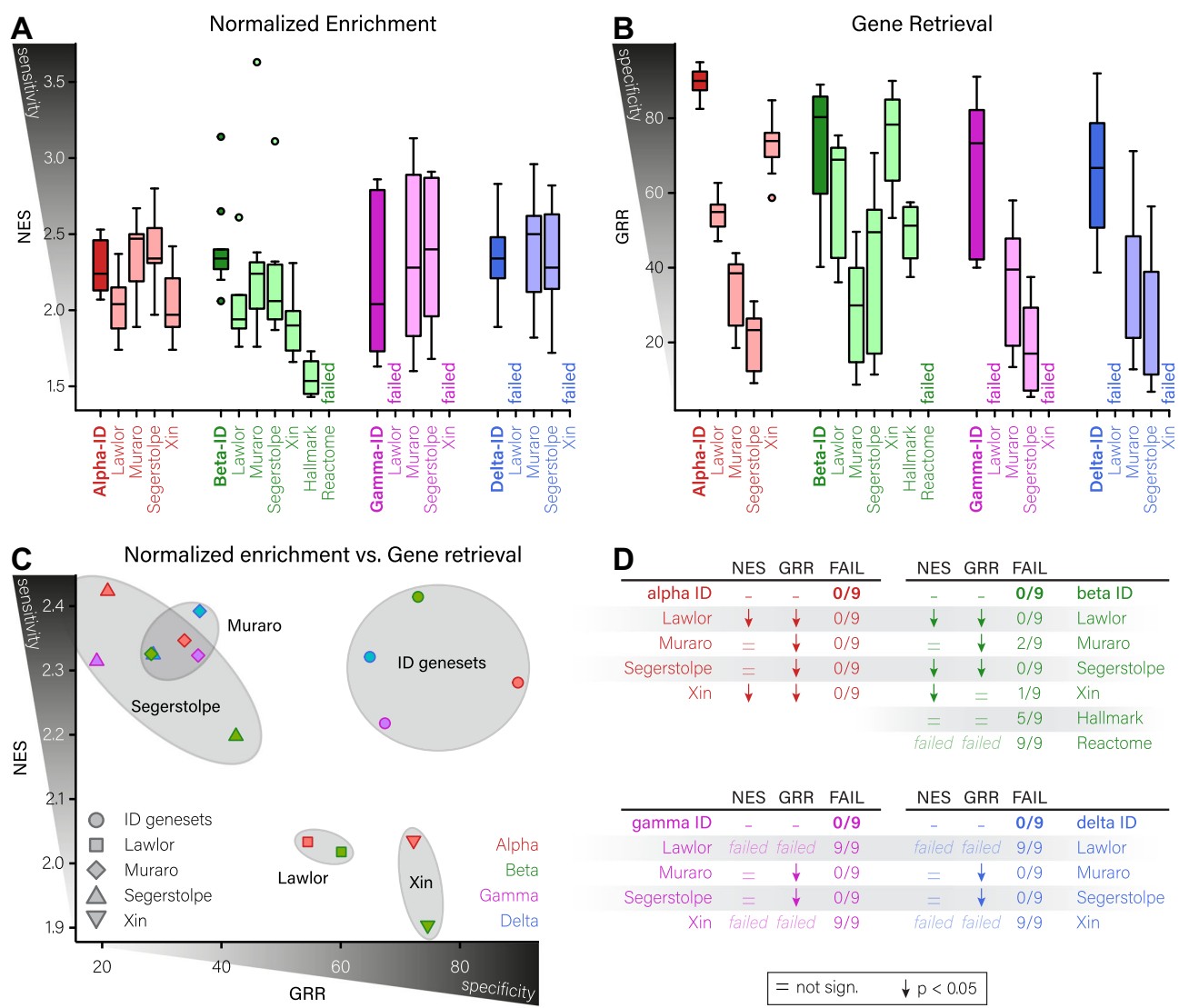

**Fig. 4 Evaluation of identity genesets reveals superior on- and off-target scoring compared to previously published ID lists. A**, **B** Box plots of normalized enrichment (sensitivity; **A**) and Gene Retrieval Rate (specificity; **B**) for our islet cell type-specific ID genesets compared to previously published ID lists, using three independent datasets for evaluation ($n = 9$ per condition). Box plots are colour-coded per cell type (α-, β-, γ- and δ-cells in red, green, magenta and blue, respectively), and distribution follows standard boxplot formatting as min-Q1-median-Q3-max, with individual dots marking outliers. Darker colours indicate our ID genesets. **C** Scatterplot indicating the relation between NES/sensitivity and GRR/specificity for the different ID genesets. Larger genesets from the Muraro and Segerstolpe datasets score well on sensitivity at the expense of GRR/specificity. The smaller genesets from the Lawlor and Xin datasets are more specific, at the expense of sensitivity. In our ID genesets, we have managed to optimize the trade-off between sensitivity and specificity. Dots are colour coded per cell type (α-cells in red, β-cells in green, γ-cells in magenta, δ-cells in blue) and shaped per dataset. **D** statistics for on-target, off-target and evaluation metric scoring. Comparisons were made using Wilcoxon signed-rank test; = indicates no significant difference compared to our ID genesets, ↓ indicates a significantly lower score compared to our ID genesets, 'failed' indicates all analyses for this geneset failed. NES normalized enrichment score, GRR gene retrieval rate, FAIL number of geneset analyses that did not produce output. Source data are provided as a source data file.

x2, both single-cell RNA-seq). The core differences between these protocols were the replacement of KGF in protocol x1 with LDN193189 during culture stages 3 and 4, and the addition of RA during stage 4 in protocol x2, which ultimately resulted in a larger fraction of α-cells in the x2 protocol compared to x1[55]. As these culture protocols mimic embryonic development, we aimed at comparing identity changes in a similar way as in the pancreas development dataset. To recapitulate this, we linked *PDX1* expressing cells at stage 3 with the ductal epithelium during embryonic development, then calculated progression towards first the *NEUROG3*, then the *FEV* expressing populations in stage 4. After that, we assumed the bifurcation as we see it during pancreas development, and calculated for α-cell differentiation

the progression towards the SC-α-cells in stages 5 and 6, while for β-cell differentiation we calculated progression towards the SC-β-cells in stages 5 and 6 (Fig. 5B, S8A). We found that in both protocols, α-cell identity was increased in cells progressing from stage 4 *FEV*+ cells towards SC alpha cells in stage 5, and again in SC alpha cells progressing from stage 5 to stage 6. Also, we found an increase in β-cell identity for cells progressing from stage 4 *FEV*+ to stage 5 SC beta cells, and for SC beta cells progressing from stage 5 to stage 6, in both protocols. Interestingly, the enrichment for α-cell identity was more pronounced in protocol x2, while the enrichment for β-cell identity was more pronounced in protocol x1. Thus, the NES using our genesets correlate with the fact that more α-cells were obtained in the x2 protocol as reported

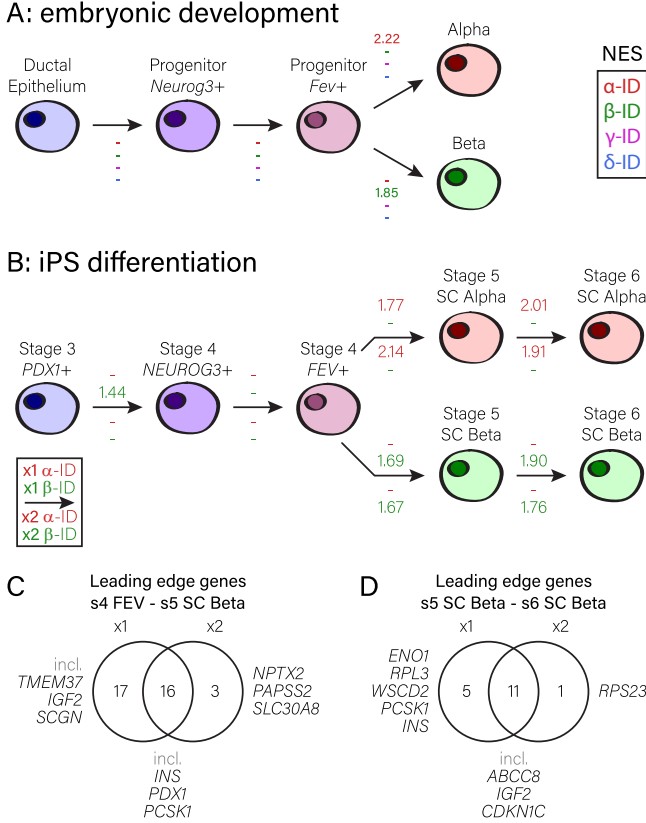

**A: embryonic development**

**B: iPS differentiation**

**Fig. 5 Assessing plasticity in α- and β-cell identity during embryonic development, pluripotent cell differentiation, cell type interconversion and in diabetes.** Changes in α-, β-, γ- and δ-cell identity were measured between states during murine embryonic pancreas development (**A**) and in different human ES/iPS cell differentiation protocols (x1 and x2; **B**). Arrows indicate progression. Values in red, green, magenta and blue indicate normalized enrichment scores from GSEA for α-ID, β-ID, γ-ID and δ-ID genesets, where positive values indicate a gain in identity and negative values indicate a loss of identity; scores indicated by – are not significant (FDR higher than 0.05). **C**, **D** Venn diagrams indicating differences in leading-edge genes (genes responsible for the correlation between a geneset and the observed increase in β-cell identity) between the x1 and x2 protocols. Differences in leading-edge genes from stage 4 *FEV* expressing cells to stage 5 SC-β cells (**C**) and from stage 5 to stage 6 SC-β cells (**D**).

in the original manuscript, and suggests they become more α-like. In contrast, the x1 protocol yielded relatively more β-cells, which we found to have a more pronounced β-like phenotype as well. Also, we found that γ- and δ-cell identities were not strongly affected in either protocol. We only observed a loss of γ-cell identity for cells progressing towards stage 5 SC β-cells in the x1 protocol (Fig. S8A). Looking at leading-edge genes (the genes detected by GSEA to be responsible for the correlation between the geneset and the observed identity changes; Fig. S8B–F), we observe several key differences between protocols x1 and x2: there seem to be more identity genes regulated in the x1 protocol compared to x2, and particularly striking is the observation that from stage 5 to stage 6 SC β-cells, key markers like *INS* and *PCSK1* are only regulated in the x1 protocol (Fig. 5C, D).

To further unravel which mechanisms play a role in the differentiation of ES/iPS cells into β-like cells, we investigated which genes in our identity genesets were regulated during the final stages of the differentiation protocol (from *FEV*+ precursors to first stage 5, then stage 6 SC-β cells), in both the x1 and x2 protocol. For this, we distinguished three sets of

regulated genes: (1) the leading-edge genes as indicated by GSEA, which support the differentiation toward the desired phenotype, (2) downregulated genes, which belong to the geneset of the objective cell-type, but are thus associated with a less differentiated, or incomplete phenotype, and (3) genes belonging to genesets of the other islet cell types in the leading-edge region of the analysis, which may indicate the acquisition of traits of alternative cell types in the differentiation protocol (Fig S8B). Regulated genes from categories 2 and 3 may indicate an incorrect or aberrant modulation of identity genes for the acquisition of the desired phenotype and thus can be used as quality markers to optimize the cells during final differentiation protocol steps.

Between both protocols, many key identity genes were correctly expressed within the different stages of β-cell maturation, like *INS, PCSK1, IGF2, ABCC8, SURF4, PDX1, MAFB* and *HADH*. On the other hand, cells expressed a few genes associated to other islet cell types (mostly with α-cells, in both protocols), like *GCG, TTR, LOXL4, SLC7A2, TMEM176B* and *ISL1*. Yet specific differences between the protocols could also be found. Beyond the leading-edge genes mentioned earlier, we found *PAX6* misexpression in the x1 protocol, whereas the x2 protocol misexpressed inhibitor of differentiation *ID1*. We also recognized the unexpected downregulation of *UCHL1, DLK1* and *LDHB* in the x1 protocol, while *GNAS, ALCAM* and *PLCXD3* were downregulated in the x2 protocol. In conclusion, we feel that the inclusion in our carefully annotated genesets of genes that are not known in the context of β-cell identity would warrant a closer inspection for their roles (Fig. S8C–F).

Next, we investigated if our genesets were also suitable to quantify cell type interconversion in adult islet cells. For that purpose, we downloaded bulk RNA sequencing data describing the conversion of adult human α-cells[47]. The original manuscript described a gain in β-cell identity by aggregating α-cells into 3D pseudo-islets (αGFP). This β-like signature was further enforced by overexpressing transcription factors Pdx1 and Mafa (αPM), resulting in glucose-sensing insulin-secreting pseudo-islets. In line with the original findings, β-cell identity was enforced after aggregating sorted α-cells and even more pronounced under αPM conditions (Fig. S9A). Negative enrichment was not observed using the α-ID geneset, which supports the initial conclusion that converted human α-cells retain a strong α-cell identity while acquiring the ability of glucose-dependent insulin secretion. The hybrid phenotype of these αPM cells can be observed by comparing them with native sorted β-cells. Here, β-cells have a strongly enriched β-cell identity, while α-cell identity is negatively impacted (Fig. S9A). By recapitulating the findings of the original manuscript, we validate that these genesets are well suited to evaluate adult islet cell type plasticity.

Lastly, we assessed if our ID genesets could reveal changes in α- and β-cell identity in disease conditions by comparing cells from NDM and T2DM donors. To this effect, we generated an integrated dataset with α- and β-cells from four of the original datasets that comprise both NDM and T2DM donors[17,19–21]. After data integration, we found that cells clustered based on cell type, not diabetes status or dataset (Fig. S9B), suggesting that the impact of T2DM does not strongly affect cellular identity. We also observed that there were many more differentially expressed genes between α- and β-cells in both ND and T2D conditions, compared to differentially expressed genes between ND and T2D cells in both α- and β-cells (Fig. S9C). When comparing NDM α- or β-cells to their T2DM counterparts, we found none of the ID genesets regulated in either cell type (Fig. S9D). This suggests that functional defects of α- and β-cells in T2DM do not seem to impact significantly on the expression of specific identity markers that comprise our genesets.

In summary, our α-ID and β-ID genesets accurately benchmark identity changes toward the α- and β-cell fates, during both embryonic development and in vitro differentiation. Likewise, they are suited to capture identity changes in adult islet cells.

## Discussion

Here we present a meta-analysis of seven independent scRNA-seq datasets[15–21] that were used to generate robust lists of identity genes for the different cell types constituting human pancreatic islets: α-, β-, γ- and δ-cells. From these lists of identity genes, we were able to generate highly sensitive and specific genesets that accurately define the cellular identity of the different pancreatic islet cell types. These final genesets represent additional tools that will help evaluating current in vitro cell differentiation protocols as well as adult islet cell type interconversion processes, amongst others.

In our primary analysis, we provide a list of genes that can be marked as identity genes for all of the islet cell types. The confidence in these genes is increased with a higher number of integrated analyses a gene is found in. This confidence was further strengthened by a deep literature review on the top regulated genes, as well as orthologous validation of specific markers for α- and β-cells. This still leaves room for interpretation of how important any given gene is in the context of cellular identity, but also sketches some interesting nuances for specific genes. For example, *ARX*, a critical transcription factor in the context of α-cell identity, was found to be strongly regulated in α-cells compared to β- and δ-cells, but not against γ-cells, which is in line with developmental findings where α- and γ-cells both develop from Arx-expressing endocrine progenitors[51]. Equally interesting, we were able to define cell surface markers that were specifically regulated between α- and β-cells, like serine peptidase inhibitor *SERPINA1* in α-cells and serine peptidase *PRSS23* in β-cells. The combination of serine peptidase and their respective inhibitors are known to play a role in ECM degradation[56] and may be involved in paracrine signalling between α- and β-cells[57].

In order to generate genesets accurately defining the different islet cell types, we tried to find the delicate balance between including as many meaningful genes as possible to provide robust enrichment scores, while keeping the genesets as concise as possible to prevent the inclusion of the more exotic, poorly predictive genes. To our knowledge, this is the first attempt to develop a methodology to make a quantifiable distinction between different genesets. As proxy for geneset sensitivity, we propose to use the normalized enrichment score, a standard output of GSEA, that explains the correlation between a geneset and an observed phenotype. To measure geneset specificity, instead, we propose to use the gene retrieval rate from GSEA analysis. This GRR value represents the proportion of genes in a geneset that were retrieved during analysis, and thus allows to specify how many genes in a geneset can be considered to be meaningful. Using this rationale, and in comparison with the genesets from the original manuscripts, we succeeded in generating genesets with optimal sensitivity/specificity balance. We also consider, in this light, the poor results observed using the widely used and appreciated Hallmark (Pancreas beta cells) geneset, where inclusion of aberrant genes like *GCG*, *SST* and *NEUROG3* can only interfere with correct recognition of the β-cell phenotype, and scored very poorly in our hands.

Geneset analyses represent a valuable approach to assess the cell type identities in various biological systems. We thus applied our genesets to evaluate processes like reprogramming through cell type interconversion, in vivo and in vitro differentiation conditions and the impact of disease (type 2 diabetes) on cellular identity. We assessed how α- and β-cell identities were affected in these conditions by retrieving transcriptomic data from recently published studies. Importantly, our human islet cell genesets can be used in a broad variety of conditions: They perform equally well in both single-cell and bulk RNA sequencing data, but also on mouse data using ortholog genes for the genesets. Geneset analyses confirmed previous results of our laboratory showing that human adult α-cells ectopically expressing Pdx1 and MafA exhibit a significant gain in β-cell identity while preserving a strong α-cell signature[47,58,59].

Finally, our genesets are effective tools to evaluate in vitro ES/iPS protocols aimed at generating surrogate β-like cells. We show that our genesets represent critical evaluation tools to quantify cell differentiation outcomes. This was done by assessing the acquisition of α- or β-cell identities, and by investigating which genes from our genesets are regulated in this context. While the original manuscript emphasizes that different protocols can impact the cellular composition of the final culture stages, we here demonstrate that this goes hand in hand with the final identity acquisition: the x1 protocol has a higher ratio of β-cells, and according to our findings these cells also more strongly gain in β-cell identity.

Still, our genesets also have limitations in applicability. For instance, the analysis of α- and β-cells from non-diabetic and type 2 diabetic donors did not reveal any identity changes. Additional efforts may be necessary to generate genesets that accurately define diabetes disease progression at the islet cell identity level. Since single-cell transcriptomics does not yield many highly regulated genes in this context, alternative techniques like GWAS may help further elucidate this process[60,61]. Similarly, we did not observe any identity modulation during pancreas development for cells progressing towards *Neurog3* or *Fev* expression, which are features of islet endocrine precursors. With the recent appearance of datasets regarding human pancreas development[62], generating carefully annotated genesets that describe these early stages may be a crucial addition for better profiling in iPS differentiation protocols.

In conclusion, we report the generation of islet cell type-specific genesets defining the identity of pancreatic α-, β-, γ- and δ-cells. As exemplified above, these genesets represent valuable tools for the islet biology/diabetes fields. We provide these genesets as resources through the Molecular Signatures Database (www.gsea-msigdb.org)[63], and a web-based application directly accessible through the website of the Human Islet Research Network (https://rapps.hirnetwork.org/scPancMeta). Our γδε−enriched dataset can be accessed through the Gene Expression Omnibus (GSE150724).

## Methods

**Nomenclature regarding the two Xin datasets.** Since we use two datasets from the Gromada group, where Y. Xin is the first author, we will use the following naming. The dataset generated using SMARTseq technology, published in 2016[21], will be referred to as Xin, while the dataset generated using 10x genomics technology, published in 2018[53], will be referred to as Xin2.

**Downloading of single-cell RNA sequencing data.** Single-read RNA sequencing data (Lawlor et al. (SRP075970 [https://www.ncbi.nlm.nih.gov/geo/query/acc.cgi?acc=GSE86473])[17], Segerstolpe et al. (ERP017126 [https://www.omicsdi.org/dataset/arrayexpress-repository/E-MTAB-5061])[19], Wang et al. (SRP076307 [https://www.ncbi.nlm.nih.gov/geo/query/acc.cgi?acc=GSE83139]) and Xin et al. (SRP075377 [https://www.ncbi.nlm.nih.gov/geo/query/acc.cgi?acc=GSE81608])[21] datasets) were downloaded as SRA files from the NCBI sequence read archive, and then transformed to fastq files using fastq-dump version 2.8.2. Paired-end RNA sequencing data (Enge et al. (PRJNA322355 [https://www.ncbi.nlm.nih.gov/geo/query/acc.cgi?acc=GSE81547])[16], Baron et al. (PRJNA328774 [https://www.ncbi.nlm.nih.gov/geo/query/acc.cgi?acc=GSE84133])[15], and Muraro et al. (PRJNA337935 [https://www.ncbi.nlm.nih.gov/geo/query/acc.cgi?acc=GSE85241])[18] datasets) were downloaded as fastq files from the European Nucleotide Archive.

**Demultiplexing of UMI datasets**. UMI based datasets (Baron and Muraro) were demultiplexed into separate fastq files, each corresponding to a single cell. For the Muraro dataset, demultiplexing was performed using the paired-end_barcode_splitter python script (https://github.com/joel-tuberosa/paired-end_barcode_splitter) implementing the FASTX-Toolkit (http://hannonlab.cshl.edu/fastx_toolkit/) version 0.0.14. This produced pairs of files, each corresponding to a single cell. Subsequently, UMI sequences were retrieved from the fastq files, and N bases were replaced with a random base. For the Baron dataset, the dropTag function and the indrop_v1_2.xml config file of the dropEst pipeline (https://github.com/hms-dbmi/dropEst) version 0.7.1[64] were used to append cell barcode and UMI sequences to the biological read names. A custom R script was then used to demultiplex the biological fastq files into individual files, each corresponding to a single cell. A barcode was considered for downstream analysis if (1) it exactly matched one of the inDrop v1 barcodes, and (2) if it had at least 10,000 reads. UMI sequences were also retrieved from the fastq files, and N bases were replaced with a random base.

**Pseudoalignment of reads**. Biological reads from all datasets were pseudo-aligned to the Ensembl release 91 of the *Homo sapiens* transcriptome reference 38 (GRCh38) using kallisto version 0.43.1[22]. Before alignment, a kallisto index was built with a k-mer length of 31. Reads from SMART-Seq datasets (Enge, Lawlor, Segerstolpe, Wang and Xin datasets) were pseudo-aligned and quantified using kallisto quant. For the single-read datasets, the estimated fragment lengths and standard deviations were set to 300 ± 150 bp for the Segerstolpe, Wang and Xin datasets, and to 200 ± 100 bp for the Lawlor dataset, depending on the cDNA library generation kits they used. Transcript counts were then summarized to gene counts using tximport version 1.8.0[65]. The Baron and Muraro datasets were pseudo-aligned to the transcriptome using kallisto pseudo in --umi mode. This pseudo-alignment produces a table of counts summarizing the number of distinct UMIs aligned to each equivalence class (EC) per cell. Each EC is formed of a (or multiple) transcript(s) to which a read can align. UMI counts per EC and per cell were corrected for collisions, as described previously[66], using the formula $m = -n \ln(1 - (b-1)/n)$, where n represents the maximum expected UMI count ($4^l$, with $l$ corresponding to the length of the UMI) and $b$ is the observed number of UMI counts. Corrected UMI counts per EC and per cell were then divided by the number of transcripts per EC and then summed into gene counts per cell.

**Downloading of processed data**. The following datasets were downloaded as processed data: matrices from 6x non-diabetic and 3x type 2 diabetic donors from Fang et al. (GSE101207)[52], matrices from 12x non-diabetic donors from Xin2 et al. (GSE114297)[53], the Seurat subset Rdata object from van Gurp et al. (GSE132364)[54], matrices from stage 3-6 for protocol X1 from Veres et al. (GSE114412)[55], all samples for sorted α, sorted β, αGFP and αPM conditions from Furuyama et al. (GSE117454)[47].

**Generation of a γ-, δ- and ε-cell-enriched single-cell transcriptomics dataset**. Human pancreatic islets from anonymized deceased donors were purchased and obtained through the NIDDK's Integrated Islet Distribution Program (IIDP), which provides islets for fundamental research worldwide (NIH Grant no. DK098085). This type of investigation is outside the scope of the Swiss Human Research Act, and does not require approval by the IRB. Human islets from three independent non-diabetic donors (RRID: SAMN11633049 [https://www.ncbi.nlm.nih.gov/biosample/?term=SAMN11633049], SAMN11963659 [https://www.ncbi.nlm.nih.gov/biosample/?term=SAMN11963659], SAMN12227196 [https://www.ncbi.nlm.nih.gov/biosample/?term=SAMN12227196]) were dissociated and labelled with cell-surface antibodies as described previously[44,47,48]. Cells were sorted on a Moflo Astrios (Beckman Coulter) system. Populations gated in HIC3-2D12 vs. HIC1-2B4 plots were further gated in a CD9 vs. SSC-H plot. For γ- and ε-cell enrichment, all CD9+ cells were complemented to 15,000 cells with CD9- cells. For δ-cell enrichment, 15,000 CD9+ and SSC-H^LOW cells were sorted. The γ/ε- and δ-fraction were processed as independent experiments using the Chromium single-cell gene expression protocol v3 (10x Genomics). A median of 55,000 paired-end reads per cell were sequenced using Illumina Hiseq4000. Data were mapped against the *Homo sapiens* transcriptome reference 38 (GRCh38) using 10x Genomics Cell Ranger v3.0.2. Cells were included if they contained at least 2000 UMIs, expressed at least 1000 genes and had a percentage of mitochondrial gene counts below 15%. Cells originating from the γ/ε-fraction and δ-fraction were merged per donor before further processing.

**QC filtering and data processing**. Seurat objects were generated for individual single-cell datasets using Seurat version 2.3.4 (Baron, Enge, Fang, Lawlor, Muraro, Segerstolpe, Wang, Xin and Xin2 datasets)[7]. Cells were filtered using dataset-specific thresholds to contain a minimum number of counts or UMIs, a minimum number of expressed genes and a maximum percentage of mitochondrial counts (Table S1). Gene counts were normalized by the library size and log-transformed. For each dataset, 40 principal components (PCs) were computed based on detected variable genes using default settings, and then tested for statistical significance using JackStraw (Supplemental Table 1). UMAP dimensional reduction was calculated using only the significant PCs. Seurat objects for the γ-, δ- and ε- enriched dataset, and the Veres dataset, were created using Seurat version 3.1.0. Individual Seurat objects were generated per donor (γ-,δ- and ε- enriched dataset) or per stage

(Veres dataset). Filtering, normalization, principal component analysis and dimensional reduction were performed as described above. Data integration per donor (our dataset) or stage (Veres dataset) was then performed using the FindIntegrationAnchors function with the top 20 CCs (our dataset) or 50 CCs (Veres dataset). The IntegrateData function was run using the top 2000 variable genes between the individual objects. PCA and UMAP dimensional reduction were performed on the integrated dataset, as described above.

**Doublet cell detection**. For the γ-, δ- and ε- enriched dataset, and the Baron, Enge, Fang, Lawlor, Muraro, Segerstolpe, Veres, Wang, Xin and Xin2 datasets, doublets were detected in each dataset using two independent tools: DoubletFinder v2.0.2 and Scrublet v0.2.1[24,25]. For DoubletFinder, 25% of the total cells were generated in doublets (pN), neighbours (pK) were defined using the parameter sweep functions and 5% of all cells in every dataset were estimated as doublets (nExp). For Scrublet, the expected doublet rate was set to 5% and the call_doublets threshold was set between 0.15 and 0.2. All cells detected by either tool as a doublet were removed.

**Clustering, hormone detection and cell type allocation**. For the γ-, δ- and ε-enriched dataset, and the Baron, Enge, Fang, Lawlor, Muraro, Segerstolpe, Wang, Xin and Xin2 datasets, the FindClusters function was run for a range of resolutions with sensitivities between 0.1 and 2. For datasets generated using Seurat version 2.3.4, clustree version 0.4.0 was used to assess the clustering results and the lowest sensitivity was picked that was stable over at least 2 resolutions. A phylogenetic tree was then built, and nodes were assigned an out-of-bag (OOB) error score using a random forest approach. All clusters underneath a node with an OOB above 10% were merged. The result was considered to be the final clustering for each dataset. This option was not available for datasets analyzed using Seurat version 3.1.0, and was thus skipped. For these datasets, clustering was performed on the integrated assay.

Upregulated genes were calculated for each cluster compared to other clusters using the FindAllMarkers function in Seurat, applying the "negbinom" test to UMI based data (γ-, δ- and ε-enriched dataset, Baron, Fang, Muraro and Xin2 datasets) and the MAST test[67] to the other datasets, and using the number of genes and counts as variables to regress. For the γ-, δ- and ε-cell enriched dataset, donor origin was included as well. Only positive genes were detected with a log fold change of at least 0.5 and an adjusted p-value of at most 0.05. For each cluster, we tried to assign one of the islet cell types based on the expression of key markers, like *GCG*, *TTR*, *ARX*, *IRX2* or *PCSK2* in α-cells, *INS*, *IAPP*, *PDX1*, *MAFA*, *NKX6-1*, *ABCC8* or *PCSK1* in β-cells, *PPY*, *STMN2*, *ETV1* or *SLC6A4* in γ-cells and *SST*, *LEPR*, *BCHE*, *HHEX* and *RGS2* in δ-cells. Cluster identity for each cell was stored as meta data.

Next, cells were classified based on their expression of hormone genes *GCG*, *INS*, *PPY*, *SST* and *GHRL*. As the expression pattern for these hormones was bimodal, density plots were generated using log-normalized counts in which the local minimum between the two modes was calculated. This was considered the threshold value for that given hormone gene, and cells were classified to express this hormone if they had an expression level equal to or higher than the threshold value. This was done for each hormone gene, after which the hormone expression profile for each cell was stored as meta data.

For final cell type allocation, we primarily assigned cell identities based on clustering. Then, we checked for each cell if their hormone expression was in line with their assigned cell type. If a cell expressed a single hormone that did not match the cluster it was found in, its cell type identity was corrected based on hormone expression (e.g. if a cell clustered with α-cells but expressed only *INS*, it would be re-assigned as a β-cell). All cells that expressed more than one hormone were categorized apart as bihormonal cells. In each dataset, we thus finally ended up with seven potential identity classifications: α-cell, β-cell, γ-cell, δ-cell, ε-cell, bihormonal, or other for cells that did not express any hormone genes and did not cluster in an endocrine cluster.

**Characterization of mono- and bi-hormonal cells**. For the Baron, Enge, Lawlor, Muraro, Segerstolpe, Wang and Xin datasets, summary statistics were generated based on how many cells expressed any given (combination of) hormone(s), split per diabetes status when applicable. The percentage of mono-hormonal cells was calculated based on the total of all mono-hormonal cells. For bi-hormonal cells, the percentage was calculated based on that given combination of hormones (e.g. the percentage of *GCG*/*INS* co-expressing cells was calculated as part of all cells that expressed *GCG* and/or *INS*), split—when applicable—per diabetes status in each dataset. In order to characterize gene expression profiles of *GCG*/*INS* expressing bi-hormonal cells, we compared mono-hormonal cells with bi-hormonal cells within the α- and β-cell clusters (i.e. GCG expressing cells vs. GCG/INS co-expressing cells in the α-cell cluster, and INS expressing cells vs. GCG/INS co-expressing cells in the β-cell cluster). Differential expression was calculated using the FindMarkers function in Seurat, applying the "negbinom" test to UMI based datasets (Baron and Muraro) and the MAST test[67] to the other datasets, and using the number of genes and counts as variables to regress.

**Defining islet cell type-specific identity genes**. For the Baron, Enge, Lawlor, Muraro, Segerstolpe, Wang and Xin datasets, upregulated genes were calculated for each identity using a pairwise approach between non-diabetic α-, β-, γ- and δ-cells (e.g. α-cells vs. β-cells, α-cells vs. γ-cells and α-cells vs. δ-cells, within one dataset, to define α-cell identity). The "negbinom" test was applied on UMI based datasets (Baron and Muraro) and the MAST test[67] to the other datasets using the Find-Markers function in Seurat. The number of genes and counts were given as variables to regress. Only positive genes were detected with a log fold change of at least 0.5 and an adjusted $p$-value of at most 0.05. Average adjusted $p$-values were calculated as 10^-(mean(-log10(value 1), -log10(value 2 … -log10(value x)). Aggregated datafiles were then generated for each cell type, integrating all upregulated gene lists from all datasets, compared to all cell types, and ordering them first by the total number of analyses they were found in, then by a rank score that was calculated as -log10(average Padj) * average logFC. Metrics were also provided for individual comparisons (e.g. in α-cells: # analyses, rank scores, average logFC and average Padj compared to β-, γ- and δ-cells). For every gene, the subcellular localization of the encoded protein was determined based on the Compartments database[68]. Each gene was matched against a transcription factor database[69], and all genes with a localization to the plasma membrane or extracellular domain were cross referenced with existing cell surface protein lists[70,71]. Finally, information was added to convey which genes were included in the genesets we generated, and if there was overlap for that gene with any of the genesets for the other cell types. These tables were provided as supplemental data (Tables S4–7).

**Multiplex single-molecule fluorescent in situ hybridization**. Between 9 and 22 DNA probes were designed for each gene that cover the mRNA coding sequence, and had overhangs that were compatible with the MUSE amplification system (Arcoris bio AG, Switzerland; Supplementary data 8). Sections of formalin fixed, paraffin embedded human pancreata were kindly provided by D. Bosco (Laboratoire d'Isolement et de Transplantation Cellulaires, Dépt de Chirurgie, Geneva University Hospitals). For these experiments, sections no older than 3 months were baked and deparaffinized, then consecutively treated with hydrogen peroxide, target retrieval solution and protease solution according to the RNAscope multiplex fluorescent reagents kit v2 (323100-USM, ACD). Then, sections were washed at 40 C in 30% formamide solution with 40 U/ml murine RNAse inhibitor in 2× SSC for 3 h and then hybridized with all probes targeting all the genes that needed to be multiplexed combined in a hybridization buffer (containing 10% (w/v) dextran sulfate, 30% formamide, 200 µg/ml BSA, 1 mg/ml yeast tRNA and 40 U/ml murine RNAse inhibitor in 2× SSC) overnight at 40 C. After, cells were consecutively treated with the primary, secondary and read-out MUSE chemistry to amplify smFISH signals, appropriate to the overhangs on the probes according to the manufacturer's standardized protocol (MUSE-5, MUSE-6 and MUSE-8; Arcoris bio AG, Switzerland). Sections were counterstained with DAPI and images were acquired on a Leica TCS SPE confocal. To measure mean intensities for α- and β-cells in individual islets, ImageJ was used to threshold GCG and INS channels, then selections were created and saved as ROIs. These ROIs were overlaid on the image of the gene to measure, and mean intensity within the ROI was calculated. This was done in two donors for five islets per donor ($n = 10$ per gene total). Results were plotted as paired boxplots, and statistical analysis was performed using Wilcoxon signed-rank test, where the α- and β-regions for each islet were taken as paired data.

**Immunofluorescence stainings**. Sections of formalin-fixed, paraffin-embedded human pancreases were provided by the Network for Pancreatic Organ donors with Diabetes (nPOD) program. They were deparaffinized, and citrate-based antigen retrieval was performed in a pressure cooker at 121 C. Primary antibodies used were mouse-α-GCG (1:1000, Sigma G2654), goat-α-PPY (1:1000, Novus Biologicals NB100-1793) and rabbit-α-ARX (1:500, gift from Ken-ichirou Morohashi). The ARX signal was amplified using a donkey-α-rabbit biotin-SP antibody (1:250, Jackson Immunoresearch 711-065-152). Secondary antibodies used were donkey-a-mouse Alexa488 (1:600, Thermo Fisher A21202), donkey-a-goat Alexa568 (1:600, Thermo Fisher A11057) and streptavidin Alexa647 (1:600, Thermo Fisher S32357). Sections were counterstained with DAPI and images were acquired on a Leica TCS SPE confocal.

**Generation and evaluation of identity genesets**. Intersect levels were defined as the minimum number of differential expression analyses a gene was found in. For example, if a gene was found to be regulated in 16 out of 21 analyses, it was placed at intersect level 16. Per cell type, genesets were generated for each unique intersect level, and these genesets contained every gene that was placed at that intersect level or higher. This yielded 21 genesets per cell type, and 84 genesets in total for the 4 islet cell types we investigated here. For downstream analyses, only genesets were included that contained between 40 and 500 genes. These genesets were used as input for GSEA, using the GSEA mac app version 4.0.1 that can be downloaded from http://www.gsea-msigdb.org/gsea/downloads.jsp. In the γδε-dataset, differential expression analysis was performed as described above, with a logFC threshold of 0.1. Rank files were calculated as the -log10(Padj), where adjusted $p$-values smaller than 1e-300 were increased to 1e-300. For each cell type, there were thus 3 GSEA results for both normalized enrichment scores (NES) and the number of genes used per analysis. For on-target scoring, we took the average of the 3 NES

values. For off-target scoring, we calculated the gene retrieval rate (GRR) as the average of the percentage of genes used over the total number of available genes in each geneset. The evaluation metric (EM) was calculated by multiplying the on-target and off-target scores. If scores could not be calculated as fewer than 15, or more than 500 genes were retrieved for GSEA, EM scores for that geneset were set to 0. For each cell type, the geneset was picked with the highest EM, and these were considered the final genesets.

**Comparison between our ID genesets and previously published lists of genes**. Four of the seven manuscripts that we used for this analysis, published lists of identity genes for each of the four cell types we here investigate (Lawlor, Muraro, Segerstolpe and Xin datasets). We converted these lists to genesets in order to compare them to our final genesets. We recycled the rank files generated from the γδε-dataset above, and generated rank files for pairwise differential expression within the Fang and Xin2 dataset in the same manner as described above. Thus, during GSEA, we now obtained 9 NES, GRR and EM scores for each geneset. For NES and GRR scores, failed analyses were presented as NA, while for EM scores, failed analyses were presented as 0 (zero) scores. Statistical analysis was performed using the Wilcoxon signed-rank test.

**Processing of the van Gurp mouse embryonic pancreas dataset**. For the van Gurp dataset, a Seurat object (version 2.3.4) was offered as supplemental data. In this object, we assigned identities based on the clustering of the original manuscript[54], as follows: clusters 2, 3 and 6 were designated as ducts, cluster 5 as *Neurog3+*, cluster 4 as *Fev+*, clusters 8 and 9 as α-cells and cluster 1 and 7 as β-cells.

**Assigning of cellular identity in the Veres human iPS/ES single-cell dataset**. Cellular identities were assigned as provided by the original manuscript: *PDX1*-expressing, *NKX6.1*-expressing, *NEUROG3*-expressing, *FEV*-expressing, SC-α-cell, SC-β-cell, SC-δ-cell, exocrine cell, enterocyte, *FOXJ1*-expressing, replicating or unknown[55].

**Integration of the Baron, Muraro and our human pancreatic single-cell datasets**. Seurat objects (version 3.1.0) were generated for the Baron, Muraro and our dataset containing all cells, and normalization and variable gene selection were performed as described above. Datasets were integrated with the FindIntegrationAnchors and IntegrateData functions of Seurat 3.1.0, using 2000 anchor features and 30 CCAs. UMAP dimensional reduction and clustering were performed as described above, using the first thirty principal components. Cell type identities were assigned based on clustering, with a resolution of 0.1.

**Integration of the Lawlor, Segerstolpe, Wang and Xin human pancreatic single-cell datasets**. Seurat objects (version 2.3.4) were generated for the Lawlor, Segerstolpe, Wang and Xin datasets using only α-, β-, γ-and δ-cells. Cellular identity (α, β, γ or δ), diabetes status (non-diabetic or type 2 diabetic) and dataset of origin were annotated in the meta data. All genes that were present in the top 1000 most highly dispersed genes in at least two datasets were identified and the four datasets were then integrated using the RunMultiCCA function of Seurat version 2.3.4. The first 8 CCAs were selected for UMAP dimensional reduction after manual interpretation of the standard correlation strength of each component.

**Differential expression analysis in the van Gurp, Veres and Lawlor/Segerstolpe/Wang/Xin integrated datasets**. Differential expression analysis was performed in a pairwise manner, using the FindMarkers function in Seurat. For the van Gurp and Veres datasets, the "negbinom" test was used, while the MAST test was applied to the integrated dataset. The number of counts and genes were used as variables to regress, and the dataset of origin was included as well for the integrated dataset. Both positive and negative genes were kept with a log fold change of at least 0.1 and an adjusted p-value equal to or below 0.05.

**Annotation of genes based on subcellular localization of encoded proteins**. Genes in Fig. 5D, E and S8C–D were annotated based on the subcellular localization of their encoded proteins. These were determined primarily based on the compartments[68] database and human protein atlas[72]. In case of ambiguity, the compartments database was used as the primary source of information. Categories were compressed to nuclear, ER, mitochondrial, cytoplasmic, secreted/granular, receptor/transmembrane and extracellular.

**Processing of the Furuyama adult cell type interconversion bulk RNA sequencing dataset**. Bulk RNA sequencing data from the Furuyama dataset was processed using DESeq2 version 1.24.0[73]. After estimation of size factors and dispersion, differential expression between samples of different populations (sorted α vs. αGFP, sorted α vs. αPM and αPM vs. sorted β) was calculated in a pairwise manner using negative binomial GLM fitting. Log2 fold changes were shrunken using lfcShrink (applying the adaptive t prior shrinkage estimator from apeglm

version 1.6.0). Both positive and negative genes were detected with a log fold change of at least 0.5 and an adjusted *p*-value of at most 0.05.

**Statistical analysis**. Unless otherwise noted, comparisons between distinct samples were tested for significance using a two-sided Wilcoxon ranked sum test.

**Reporting summary**. Further information on research design is available in the Nature Research Reporting Summary linked to this article.

## Data availability

Genesets can be downloaded from MSigDB (https://www.gsea-msigdb.org/gsea/msigdb/index.jsp). All data in this manuscript can be accessed through an R-shiny based web app (https://rapps.hirnetwork.org/scPancMeta). Our γ- δ- and ε-cell enriched datasets can be accessed through the Gene Expression Omnibus (accession nr. GSE150724). The following datasets were downloaded as SRA files from the NCBI sequence read archive: Lawlor[17] (SRP075970 [https://www.ncbi.nlm.nih.gov/geo/query/acc.cgi?acc=GSE86473]), Segerstolpe[19] (ERP017126 [https://www.omicsdi.org/dataset/arrayexpress-repository/E-MTAB-5061]), Wang[20] (SRP076307 [https://www.ebi.ac.uk/ena/browser/view/PRJNA325005?show=reads]) and Xin[21] (SRP075377 [https://www.ebi.ac.uk/ena/browser/view/PRJNA322072?show=reads]). The Baron[15] (PRJNA328774 [https://www.ebi.ac.uk/ena/browser/view/PRJNA328774?show=reads]), Enge[16] (PRJNA322355 [https://www.ebi.ac.uk/ena/browser/view/PRJNA322355?show=reads]) and Muraro[18] (PRJNA337935 [https://www.ebi.ac.uk/ena/browser/view/PRJNA337935?show=reads]) datasets were downloaded as fastq files from the European Nucleotide Archive. These datasets were downloaded as processed data: Fang[52] (GSE101207), Xin[53] (GSE114297), van Gurp[54] (GSE132364), Veres[55] (GSE114412) and Furuyama[47] (GSE117454). Source data are provided with this paper.

## Code availability

All codes used to generate data in this manuscript are available upon justified request. Code used to share data is available on GitLab (https://gitlab.com/hirn-apps/scpancmeta).

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

## Acknowledgements

We are grateful to Klaus Kaestner, for carefully reading and commenting on the manuscript. We would like to thank Joël Tuberosa for developing the paired-end_barcode_splitter python script and Yvan Wenger for his helpful advice on data pre-processing. We also would like to thank Dave Ko and Sherland Fuertez for their assistance in setting up the technical infrastructure to host the scPancMeta application. GSEA plots were generated based on the replotGSEA tool developed by Thomas Kuilman. The tool can be found here: https://github.com/PeeperLab/Rtoolbox. This research was performed with the support of the Network for Pancreatic Organ donors with Diabetes (nPOD; RRID:SCR_014641), a collaborative type 1 diabetes research project sponsored by JDRF (nPOD: 5-SRA-2018-557-Q-R) and The Leona M. & Harry B. Helmsley Charitable Trust (Grant#2018PG-T1D053). The content and views expressed are the responsibility of the authors and do not necessarily reflect the official view of nPOD. Organ Procurement Organizations (OPO) partnering with nPOD to provide research resources are listed at http://www.jdrfnpod.org/for-partners/npod-partners/. This work was funded with grants from the Swiss National Science Foundation (310030B_173319 and 310030_192496 to P.L.H., and 310030_189153 to I.R.), the Fondation Aclon (to P.L.H.), and the European Research Council (Advanced Grant "Merlin", #884449, to P.L.H.). J.S.K. was supported by the City of Hope (intramural) and NIH/NIDDK (U24DK104162).

## Author contributions

Conceptualization L.v.G., D.O., F.T., P.L.H.; software L.v.G., L.F., A.V.U., J.S.K.; formal analysis L.v.G.; investigation: L.v.G., K.F., E.B.T.; visualization: L.v.G., A.V.U., J.S.K.; writing - original draft: L.v.G., F.T.; writing - review and editing L.F., D.O., K.F., E.B.T., A.V.U., J.S.K., I.R., P.L.H.

## Competing interests

The authors declare no competing interests.
