## [Peer Review File · Nature Communications]

Editorial Note: This manuscript has been previously reviewed at another journal that is not operating a transparent peer review scheme. This document only contains reviewer comments and rebuttal letters for versions considered at *Nature Communications* .

REVIEWER COMMENTS

Reviewer #2 (Remarks to the Author):

The authors' thoughtful responses and additional analyses are appreciated and add some important validation and context to the signature gene sets they identified in this thorough meta-analysis. I appreciate their clearer, integrated representation of the NES and GRR, particularly the revised Figure 4C, contextualizing their findings of the comparative gene set analyses and the revised analyses of the differentiation and trans-differentiation experiments and data sets. As such, my enthusiasm for the technical merits of this manuscript has grown, and I am happy to recommend it for publication pending the following few questions and recommendations:

- 1) In revised Figure 5, it appears that all islet cell type gene sets were tested in 5A embryonic development and initial 3 steps of 5B iPS differentiation, but not the subsequent steps. Were the delta and gamma gene sets indeed tested for enrichment or depletion in the 5B iPS differentiation? What were the results of those analyses? To be comprehensive and complete, the results and interpretation of these analyses should be included.
- 2) Page 12, lines 379-383: I'm not convinced that the NES qualitatively indicates that the x2 protocol yields a stronger alpha-like phenotype or can distinguish changes in the relative number or composition of the alpha cells from the phenotype strength of each cell when these data are derived from bulk transcriptome measurements. Single cell would seem to be necessary to definitively conclude if these stronger NES signatures are just due to more high quality alpha cells being present in the x2 protocol cell composition vs. 'better' alpha cells. This comment should be omitted, tempered, or modified. It seems the NES is compatible with more and/or better alpha cells in the x2 protocol.

Minor comments:

- 1) There are multiple places in the manuscript where nomenclature conventions are not quite correct--for example, gene names need to be italicized, protein names not, and human vs. mouse genes/proteins should be distinguished using standard conventions of all capital letters for human genes/proteins and first capital letter for mouse genes/proteins.
- 2) Commas in the manuscript text appear to be mistakenly replaced with apostrophes in places, e.g., page 7, paragraph starting "The new dataset was generated using 10X Genomics...."

Reviewer #2 (Remarks to the Author):

The authors' thoughtful responses and additional analyses are appreciated and add some important validation and context to the signature gene sets they identified in this thorough meta-analysis. I appreciate their clearer, integrated representation of the NES and GRR, particularly the revised Figure 4C, contextualizing their findings of the comparative gene set analyses and the revised analyses of the differentiation and trans-differentiation experiments and data sets. As such, my enthusiasm for the technical merits of this manuscript has grown, and I am happy to recommend it for publication pending the following few questions and recommendations:

We appreciate the continued candid opinion of the reviewer, as we feel it has considerably helped us improve the quality of this manuscript, for which we are grateful.

- 1) In revised Figure 5, it appears that all islet cell type gene sets were tested in 5A embryonic development and initial 3 steps of 5B iPS differentiation, but not the subsequent steps. Were the delta and gamma gene sets indeed tested for enrichment or depletion in the 5B iPS differentiation? What were the results of those analyses? To be comprehensive and complete, the results and interpretation of these analyses should be included.

We understand this request, as this may seem inconsistent between the different experiments. We had already performed the analyses; they were included in a previous revision of the manuscript, but we decided to remove them for clarity and focus in the figures, emphasizing the differences between alpha and beta cells in the different culture protocols. The information for gamma and delta cells has now been added back, in supplemental figure S8A.

- 2) Page 12, lines 379-383: I'm not convinced that the NES qualitatively indicates that the x2 protocol yields a stronger alpha-like phenotype or can distinguish changes in the relative number or composition of the alpha cells from the phenotype strength of each cell when these data are derived from bulk transcriptome measurements. Single cell would seem to be necessary to definitively conclude if these stronger NES signatures are just due to more high quality alpha cells being present in the x2 protocol cell composition vs. 'better' alpha cells. This comment should be omitted, tempered, or modified. It seems the NES is compatible with more and/or better alpha cells in the x2 protocol.

We apologize for the lack of clarity. Both the x1 and x2 data are based on single cell transcriptomics, which we have now tried to make clearer in the text. Also, we blunted the statement from "the NES also indicates that, qualitatively, these cells also acquire a stronger α -like phenotype" to "suggests they become more α -like".

Minor comments:

- 1) There are multiple places in the manuscript where nomenclature conventions are not quite correct--for example, gene names need to be italicized, protein names not, and human vs. mouse genes/proteins should be distinguished using standard conventions of all capital letters for human genes/proteins and first capital letter for mouse genes/proteins.

We have thoroughly gone over the text to correct this where necessary.

- 2) Commas in the manuscript text appear to be mistakenly replaced with apostrophes in places, e.g., page 7, paragraph starting "The new dataset was generated using 10X Genomics....

We thank the reviewer for noting this. We fear this is an artefact created by pdf conversion online, as these mistakes do not appear in our local version, and will be mindful of this moving forward.

REVIEWERS' COMMENTS